# Sample Efficient Active Learning of Causal Trees

**Kristjan Greenewald**
IBM Research
MIT-IBM Watson AI Lab
kristjan.h.greenewald@ibm.com

**Dmitriy Katz**
IBM Research
MIT-IBM Watson AI Lab
dkatzrog@us.ibm.com

**Karthikeyan Shanmugam**
IBM Research
MIT-IBM Watson AI Lab
karthikeyan.shanmugam2@ibm.com

**Sara Magliacane**
IBM Research
MIT-IBM Watson AI Lab
sara.magliacane@ibm.com

**Murat Kocaoglu**
IBM Research
MIT-IBM Watson AI Lab
murat@ibm.com

**Enric Boix-Adserà**
MIT
MIT-IBM Watson AI Lab
eboix@mit.edu

**Guy Bresler**
MIT
MIT-IBM Watson AI Lab
guy@mit.edu

## Abstract

We consider the problem of experimental design for learning causal graphs that have a tree structure. We propose an adaptive framework that determines the next intervention based on a Bayesian prior updated with the outcomes of previous experiments, focusing on the setting where observational data is cheap (assumed infinite) and interventional data is expensive. While information greedy approaches are popular in active learning, we show that in this setting they can be exponentially suboptimal (in the number of interventions required), and instead propose an algorithm that exploits graph structure in the form of a centrality measure. If each intervention yields a very large data sample, we show that the algorithm requires a number of interventions less than or equal to a factor of 2 times the minimum achievable number. We show that the algorithm and the associated theory can be adapted to the setting where each performed intervention yields finitely many samples. Several extensions are also presented, to the case where a specified set of nodes cannot be intervened on, to the case where $K$ interventions are scheduled at once, and to the fully adaptive case where each experiment yields only one sample. In the case of finite interventional data, through simulated experiments we show that our algorithms outperform different adaptive baseline algorithms.

## 1 Introduction

Causal discovery from observational and interventional data is a fundamental problem and prevalent in multiple areas of science and engineering (Pearl, 2009; Spirtes et al., 2000; Peters et al., 2017). Learning the underlying causal mechanisms is essential for policy design. Technological advancements in the recent decades have paved the way for the collection of abundant amounts of observational data, i.e., data collected *without perturbing the underlying causal mechanisms*. However, observational data is generally not sufficient for drawing causal conclusions and interventional data, i.e., data collected after a perturbation in the system, may be needed. Therefore, many recent methods propose to exploit both observational and interventional data (Triantafillou & Tsamardinos, 2015; Hyttinen et al., 2014; Peters et al., 2016; Zhang et al., 2017; Magliacane et al., 2016).

In the literature, there is growing interest in algorithms for intervention (experimental) design to learn causal graphs (Hyttinen et al., 2013; Shanmugam et al., 2015; Kocaoglu et al., 2017; Lindgren et al., 2018). These algorithms recommend the next experiment to perform to the practitioner, which is a perfect do($\mathbf{X}$) intervention Pearl (2009) on a set of intervention targets $\mathbf{X}$. Moreover, they provide guarantees that interventional data from these experiments are sufficient to recover the underlying causal graph in the minimum number of experiments. A *non-adaptive* intervention design is determined a priori before any of the interventions are performed. We focus on the *adaptive* intervention design setting, which determines the next experiment after collecting and processing the information collected from all experiments up until that point.

In many real-world settings, the collection of interventional data is much more difficult and costly than that of its observational counterpart. For example, in many medical settings there is plenty of observational clinical data, while randomized controlled trials are expensive to organize. Therefore it is generally desirable in practice to use as few interventional samples as possible. However, most existing work assumes a perfect conditional independence oracle which is only true when a very large number of samples are available from each experiment. In this work, we focus on removing this constraint: we assume that infinitely many observational samples are available, while only finitely many samples for each intervention can be obtained.

In this paper we assume causal sufficiency, i.e., the absence of latent confounders, and no selection bias. Causal inference using observational data in this setting has been extensively studied in the literature. As an example, in the PC algorithm (Spirtes et al., 2000), causal structure is recovered from conditional independence tests using the rules described by Meek (1995), which are provably complete, i.e. they recover all causal relations that can be identified from the data. The identifiable causal directions are represented as directed edges in the *essential graph*, while the non-identifiable directions are represented as undirected edges. It can be shown that each undirected component in the essential graph does not give information about the other undirected components, and therefore can be learned separately.

*Causal Forest Assumption:* In this work, we assume that each of the undirected components of the essential graph are trees (in general, the essential graph has chordal undirected components), i.e. they form a forest. Under this assumption, the graph can be decomposed into a set of undirected trees in which there are no unshielded colliders. Our assumption is satisfied when the original graph is bi-partite, since chordal components of bi-partite graphs are forests. Examples of bi-partite causal graphs occur in systems biology networks, e.g. gene-protein networks where genes cause protein expressions and expressed proteins block or activate other genes (Kontou et al. (2016)). Another motivation for focusing on learning orientations in the undirected tree components is that it would give insights for the general case when undirected components of the essential graph are chordal as chordal graphs are trees of cliques. In the remainder of the paper, we design algorithms for orienting each of these tree components individually, since their orientations are not informative of each other and must be determined in sequence.

We consider a Bayesian approach where we assume a prior distribution on the set of all possible causal graphs on a given undirected tree. The problem can be described as follows: given an undirected tree that does not contain any unshielded colliders, design experiments adaptively to learn the underlying causal graph with the minimum expected number of interventions. In this context, expectation is with respect to a given prior distribution over all causal graphs with the given tree as their essential graph. We propose an efficient algorithm for discovering the underlying causal structure that does not require a perfect conditional independence oracle to process interventional data. To illustrate the soundness of our approach, we first assume that infinite observational and interventional data are available and show that the average number of experiments required by our algorithm is within a multiplicative factor of 2 of the optimal algorithm. Extensions are then given to the case where some nodes cannot be intervened on, and to the case where $K$ nodes at a time are requested by the experimenter. We consider two adaptations of this theoretical result to the finite sample case, both based on obtaining a specified number of samples per intervention: (1) a simple union-bound based approach that samples each intervention until the confidence is sufficiently high to apply the noiseless algorithm, and (2) an approach inspired by the results of Emamjomeh-Zadeh et al. (2016) that obtains a small number of samples per intervention and maintains a Bayesian posterior of the root location. This last result requires $O(\log(n/\delta)/\epsilon^2)$ total interventional samples in expectation, in contrast to the result of Emamjomeh-Zadeh et al. (2016) for the structured noisy search problem, which has a $\mathcal{O}(\log(d))$ dependence for degree $d$.

## 1.1 Related Work

**Experimental Design** Hyttinen et al. (2013); Eberhardt (2007); Eberhardt et al. (2005) show that in the worst-case scenario $\mathcal{O} \log(n)$ experiments are necessary and sufficient to recover the causal graph, even when the algorithm is adaptive (Shanmugam et al., 2015). Hu et al. (2014) show that $\mathcal{O}(\log \log(n))$ randomized experiments are sufficient to learn the graph with high probability. Shanmugam et al. (2015) also propose an adaptive algorithm for the setting where at most $k$ nodes can be randomized. Kocaoglu et al. (2017) consider the problem of minimum-cost intervention design when each node is associated with an intervention cost and proposed a greedy algorithm, which gives a $(2 + \epsilon)$-approximation (Lindgren et al., 2018). Ghassami et al. (2017b) studied the problem of learning the maximum number of edges for a given number of size 1 interventions. Except Shanmugam et al. (2015), all of these works operate in the non-adaptive (offline) setting where experiments are designed before collecting interventional data. Moreover, all assume the existence of a perfect CI oracle after every intervention, which in general requires infinite experimental data. Recently Agrawal et al. (2019) introduced an experimental selection algorithm for learning a specific target function of the causal graph with a budget on the number of samples, i.e., in the fixed budget regime, whereas we work in the fixed confidence regime (Jamieson et al., 2014). Also recently, Ghassami et al. (2017a) proposed a non-adaptive intervention design to learn as much as possible about the underlying causal graph using at most $M$ experiments. A routine in their algorithm chooses a *central node* to intervene on. While we also use the concept of a central node, our learning algorithm and analysis is fully adaptive.

**Search on Structured Data** When the essential graph is a tree, learning the causal graph becomes equivalent to a structured search problem, since it is reduced to identifying the root node. Onak & Parys (2006) consider searching on trees where a query on a node outputs whether the queried node is the marked node and if not outputs the branch on which the marked node lies, which under infinite interventional data would be exactly our setup. However, they focus on minimizing the worst case performance while we focus on the average case. Jacobs et al. (2010) consider the edge query model on trees: An edge after being queried yields the direction in which the marked node lies. Their objective is to minimize the average case number of queries relative to an arbitrary prior which is known. Although several other variants of the search problem exist (Dereniowski et al., 2017; Emamjomeh-Zadeh et al., 2016; Dereniowski et al., 2018; Cicalese et al., 2010, 2014), even with extensions to the noisy case (Dereniowski et al., 2018), as far as we are aware searching with vertex queries to minimize average case performance has not been studied before. See (Dereniowski et al., 2017) for an overview of the literature.

## 2  Problem Statement

We assume that we have a collection of real-valued (possibly discrete) variables $\mathbf{X} = \{X_1, X_2, \ldots, X_n\}$, and have access to enough observational data to have determined the joint distribution $p(X_1, X_2, \ldots, X_n)$ over the $n$ variables. We further assume that the Causal Markov and faithfulness assumptions (Spirtes et al., 2000) hold, implying a one-to-one correspondence between d-separations and conditional independences. Combined with the assumption that the underlying graph is a tree, this implies that we have access to the correct undirected version of the causal graph.

Our goal is to learn a causal model over $X_1, X_2, \ldots, X_n$, where with a slight abuse of notation, we use $X_i$ to refer both to the random variable and to the associated node in the causal graph. Assume that there are no v-structures in the causal graph (otherwise the graph can be decomposed into smaller subgraphs which are non-informative about each other). The possible edge orientations following this constraint correspond to directed graphs $G_r$ with root node $R = r$ and all edges oriented away from $r$. Let $\mathcal{G} = \{G_r : r \in [n]\}$. In other words, the causal model can be specified completely by the identity of the root node $r$.

In what follows, we use the following graph-related notation. For any node $X_i$, let $N_G(i)$ be the set of neighbors of $X_i$ in the tree $G$, e.g., in Figure 1, $N_G(2) = \{X_1, X_4, X_5\}$. For a node $X_i$ and its neighbor $Y \in N_G(i)$, we write $B_G^{X_i:Y}$ to denote the set of nodes that can be reached from $Y$ when the edge between $X_i$ and $Y$ is cut from the graph. Note that node $Y$ is included in $B_G^{X_i:Y}$. We also define $B_G^{X_i:X_i} = \{X_i\}$. As an example, in Figure 1 the branches connected to $X_2$ are $B_G^{X_2:X_1} = \{X_1, X_3\}$, $B_G^{X_2:X_4} = \{X_4\}$, and $B_G^{X_2:X_5} = \{X_5, X_6\}$. We write the cardinality of a graph $G$ as $|G|$.

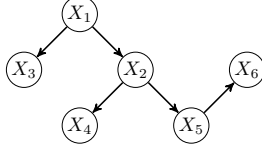

Figure 1: Graph notation example.

Our focus in this paper is to apply active learning approaches to adaptively and sequentially choose a series of interventions to best determine the causal graph (from among the set $\mathcal{G}$). For this paper, we assume interventions are single target perfect interventions, i.e., they take the form of the experimenter setting the value of some chosen node $X_i$. At each time $t$ the algorithm chooses to intervene at node $i_t$. It observes a sample $\mathbf{X}^{(t)} \sim \mathbb{P}(\cdot|\mathsf{do}(X_{i_t} = x_{i_t}))$ for some $x_{i_t}$. The algorithm runtime until the root $r$ (and the corresponding causal model $G_r$) is identified with some desired confidence $1 - \delta$ could be random or deterministic.

Given an interventional sample at node $X_i$, the posterior update contained therein can be computed via the following lemma, which is proved in Appendix B. The time index is omitted for simplicity.

**Lemma 1.** *Given an interventional sample* $\mathbf{x}$ *from* $\mathbb{P}(\mathbf{X}|\mathsf{do}(X_i = 1))$*, collected after we intervened on* $X_i$ *by setting it to* $x = 1$[1]*, the posterior update for the probability that the root is in the branch* $B_G^{X_i:Y}$*, for all* $Y \in N_G(i) \cup X_i$*, is given by*

$$\forall X_a \in B_G^{X_i:Y}, \mathbb{P}(R = X_a|\mathbf{X} = \mathbf{x}, \mathsf{do}(X_i = 1)) \propto \begin{cases} \mathbb{P}(R = X_a)\frac{\mathbb{P}(Y=y)}{\mathbb{P}(Y=y|X_i=1)} & Y \in N_G(i) \\ \mathbb{P}(R = X_a) & Y = X_i, \end{cases}$$

*where the proportionality constant does not depend on* $Y$ *and* $y$ *is the observed value of* $Y$*.*

This result implies that the only relevant interventional values are those of the neighbors $N_G(i)$ of the intervened node $X_i$. This is a critical observation that informs the development of our approach.

We will also consider the simpler setting where given the choice of a node $X_i$ on which to intervene, the experimenter returns a large number of interventional experiments on that node (assumed to be infinite). In this case, based on Lemma 1, an intervention acts to collapse the posterior distribution onto either $X_i$ (if it is the true root) or one of the adjacent branches $B_G^{X_i:Y}$ for some $Y \in N_G(i)$ a neighbor of $X_i$. We call this setting the "noiseless" setting, and use it as a starting point for the development of approaches for the more general setting.

## 2.1 Suboptimality of naïve algorithms

**Nonadaptive (without active learning feedback).** In a non-iterative setting where the outcome of the interventions are not observed until all experiments are complete, any algorithm that wishes to find the root node must take at least $O(n/(d + 1))$ interventions in expectation (under a uniform prior), where $d$ is the largest degree in the graph. This follows since each intervention only provides information about the $d + 1$ possible directions of the root from the intervened node. For bounded $d$, this is exponentially (in the number of interventions) worse than our bound for our adaptive central node algorithm.

**Information greedy algorithm is exponentially suboptimal.** An *information greedy* algorithm is one that intervenes at the node $X_i$ that in expectation reduces the entropy of the posterior on $R$ the most. Several works have proposed this approach, including Ness et al. (2017) who applied it to the intervention design setting. While attractive from an intuitive standpoint, this counterexample shows it can be exponentially suboptimal. Consider the graph in Figure 2 for parameter $K = 3$. Construct a 3-ary tree (each non-leaf node has degree 3) of minimimum depth with $K$ leaf nodes $\ell_i$, $i = 1, \dots, K$. At each of these leaf nodes, draw edges to $\lceil e^{4K} \rceil$ new nodes (where $\lceil \cdot \rceil$ denotes rounding up to the nearest integer), which become the new leaf nodes of the finished graph. Suppose that the true causal graph corresponds to this skeleton, with the directions of the edges emanating away from a root node.

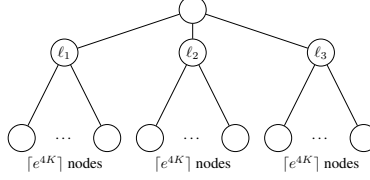

Figure 2: Counterexample for information greedy algorithm, shown for $K = 3$. The optimal algorithm can find the root in $\lceil \log_2 K \rceil + 1$ interventions by a top-down approach, while the information greedy algorithm intervenes on the $\ell_k$ ($k = 1, \ldots, K$) nodes, taking at least $K/2$ steps in expectation.

Suppose further that the unknown root note has a uniform prior distribution over all nodes in the graph. Let $n$ be the number of nodes in this example, observe that $n \leq (K+1)\lceil e^{4K} \rceil$.

Considering the "noiseless" setting where each intervention is performed infinitely many times, it is clear that the optimal algorithm can identify this root node in at most $\lceil \log_2 K \rceil + 1$ interventions. To see this, observe that one can first intervene at the center node of the graph. This gives the direction of the root from this node, so intervene at the adjacent node in that direction. Repeat this process until the root node is identified. Since the depth of the tree is $\lceil \log_2 K \rceil + 2$, this algorithm is guaranteed to find the root node by $\lceil \log_2 K \rceil + 1$ interventions.

For the information greedy algorithm, we have the following bound proved in Appendix D:

**Proposition 1.** *For the above counterexample, the information greedy algorithm will choose the nodes $\ell_i$ before any others, hence it takes a number of interventions with expected value at least $K/2$.*

This implies that the information greedy algorithm is exponentially suboptimal in this scenario with respect to the number of interventions required. Note that information greedy will also be at least nearly exponentially suboptimal in the noisy setting since the above noiseless-optimal algorithm can be extended to the noisy case via repeating interventions at each node a number of times logarithmic in the size of the graph to recover the graph with high probability.

# 3 Central node algorithm and variants

Consider the following algorithm. At time $t$, there is a *prior* distribution $p_t(R = r)$ over the nodes of the tree which is the posterior probability each node is the root given the intervention history up to but not including time $t$. The *posterior* distribution at time $t$ is formed by updating the prior $p_t(R = r)$ with the observed data $\mathbf{X} = \mathbf{x}$ to form $q_t(R = r) = \mathbb{P}(R = r | \mathbf{X} = \mathbf{x})$. Note that the posterior at time $t$ becomes the prior at time $t + 1$, i.e. $p_{t+1}(R = r) = q_t(R = r)$. We call a node $v_c$ a *central node* if it divides the tree into a set of undirected trees, each with total posterior probability less than $\frac{1}{2}$. Specifically, we have the following definition:

**Definition 1.** *A central node $v_c$ of a tree $G$ with respect to a distribution $q$ over the nodes is one for which $max_{j \in N(v_c)} q(B_G^{v_c : X_j}) \leq 1/2$. At least one such $v_c$ is guaranteed to exist (*Jordan, 1869*).*

Algorithm 4 in Appendix A gives a simple algorithm for finding such a central node with runtime linear in $n$.

We next propose the following *central node algorithm* for discovering the root node, given in Algorithm 1. At each time $t$, it intervenes on a central node with respect to the current prior and updates the prior using data from this intervention in accordance with the update in Lemma 1.

While itself a deterministic procedure, Algorithm 1 is adaptive to the outputs of the interventions and hence produces a stochastic sequence of interventions. Note that intervening on a leaf (e.g. if $q(i) > 1/2$ for some leaf node $i$) is never optimal, if the high-probability node is a leaf, one can simply intervene at its (unique by definition) neighbor and strictly improve the algorithm. We omit this special case from the algorithm and analysis for simplicity.

## 3.1 Noiseless setting: Adaptive search on a tree

We first consider the simplest case, for which we show that the central node algorithm is within a factor of 2 from the optimal. In this setting, we define the optimal algorithm to be the one that requires the smallest number of interventions, in expectation, to identify the true root. Recalling the "noiseless" setting, we start with a tree $G_0$, such that an intervention on any node $X_i$ provides the direction $u \in \{X_i\} \cup N_{G_0}(i)$ in which the root node lies, in other words the true root $r_0 \in B^{X_i : u}(G_0)$.

---

**Algorithm 1** Central Node Algorithm

---

**input** Observational tree $G_0$. Confidence parameter $\delta$.

1: $t \leftarrow 0$.
2: $q_0(i) \leftarrow \frac{1}{n}, \forall i = 1, \ldots, n$.
3: **while** $\max_i q_t(i) \leq 1 - \delta$ **do**
4:      $t \leftarrow t + 1$.
5:      Identify central node index $v_c(t)$ of $G$ with respect to $q_{t-1}$ (Algorithm 4).
6:      Intervene on node $v_c(t)$ and observe $x_1, \ldots, x_n$.
7:      Update posterior distribution $q_t$ as given in Lemma 1.
8: **end while**

**output** $\mathrm{argmax}_i q_t(i)$ as the estimated root node of $G_0$.

---

Considering this noiseless setting allows us to examine the problem in its most basic "search on a tree" form. In subsequent sections we will reintroduce various sources of uncertainty and provide strategies for handling them. Note that in this setting, having a uniform prior $p_0(i) = 1/n$ yields a uniform posterior over $G(t)$ at time $t$, hence we can compute the central nodes under a uniform distribution (e.g. $q(i) = 1/|G(t)|$). The extension to non-uniform priors is straightforward but omitted for readability. The resulting central node algorithm is shown in Algorithm 2.

---

**Algorithm 2** Central Node Algorithm (noiseless)

---

**input** Observational tree skeleton $G_0$.

1: $t \leftarrow 0, G(0) \leftarrow G_0$.
2: **while** $G(t)$ contains more than one node **do**
3:      $t \leftarrow t + 1$.
4:      Find a central node $v_c(t)$ of $G(t-1)$ under the current posterior distribution (Algorithm 4).
5:      Intervene on $v_c(t)$ and observe direction of root node $u_t \in \{v_c(t)\} \cup N_{G(t-1)}(v_c(t))$.
6:      Set $G(t) \leftarrow B_{G(t-1)}^{v_c(t):u_t}$.
7: **end while**

**output** Node remaining in $G(t)$ as the root node $r_0$ of $G_0$.

---

By the definition of a central node, we can show that Algorithm 2 converges exponentially. The question remains as to how close this rate is to that of the optimal algorithm. We prove the following theorem in Appendix E.

**Theorem 1.** *Let $G_0$ be an undirected tree for the causal discovery problem. Consider a sequence of interventions determined by Algorithm 2. Define the running time (total number of interventions) of this Algorithm to be $T_{CN}$ interventions. Let the running time of an optimal (in terms of number of interventions) algorithm that finds $r_0$ be $T_{opt}$. Then*

$$T_{CN} \leq \lceil \log_2 |G_0| \rceil, \quad \text{and moreover,} \quad \mathbb{E} T_{CN} \leq 2 \mathbb{E} T_{opt},$$

*where the expectation is with respect to the prior distribution $p_0(i) = 1/n$ over $i = 1, \ldots, n$.*

**Remark 1** (Finite sample extension). *While Algorithm 2 is written assuming that the interventions provide noiseless information (infinite sample case), it is simple to extend to the finite sample case. Specifically, if it is desired to find the correct root node with probability $1 - \delta$ for some $\delta$, by the union bound over all $n$ nodes it is sufficient to repeat the intervention on $v_c(t)$ enough times such that the probability that the returned $u_t$ is correct exceeds $1 - \frac{\delta}{n}$. It can be shown that the number of repeated interventions required to achieve this threshold is $\tilde{O}(\log(n/\delta))$. The simplicity of this finite-sample extension stands in contrast to CI testing based methods.*

*This algorithm also enjoys the practical advantage of doing batches of interventions on a single node before moving onto the next one. This will limit the number of distinct nodes for which interventions need to be run, and opens the door to running multiple interventions in parallel.*

### 3.2 Designing $K$ adaptive interventions per cycle

Consider the case where it is ideal for the experimenter to perform $K$ interventions in sequence before returning to the active learning algorithm for another set of $K$ interventions to run. In this

setting, we extend the concept of a central node to a set of $K$-central nodes that divide the graph into pieces with mass no more than $\frac{1}{K+1}$ each:

**Definition 2** ($K$-central nodes). *A set of up to $K$ nodes $v_c^k$ of a tree $G$ with respect to a distribution $q$ over the nodes is a set of nodes for which $\max_{\{j_k \in N(v_c^k)\}_{k=1}^K} q\left(\cap_{k=1}^K B_G^{v_c^k : X_{j_k}}\right) \leq \frac{1}{K+1}$.*

Similar to the central node, this set of nodes is guaranteed to exist and can be constructed in a similar fashion. Using this concept, we propose Algorithm 3, where at each step we intervene on the $K$-central nodes and update as in Algorithm 2.

---

**Algorithm 3** $K$-Central Node Algorithm (noiseless)

---

**input** Observational tree skeleton $G_0$.
 1: $t \leftarrow 0, G(0) \leftarrow G_0$.
 2: **while** $G(t)$ contains more than one node **do**
 3:     $t \leftarrow t + 1$.
 4:     Find a set of $K$-central nodes $\{v_c^k(t)\}$ of $G(t-1)$ under the uniform distribution.
 5:     Intervene on each of the $v_c^k(t)$ in sequence and for each observe the direction of root node $u_t^k \in v_c^k(t) \cup N_{G(t-1)}(v_c^k(t))$.
 6:     Set $G(t) \leftarrow \cap_{k=1}^K B_{G(t-1)}^{v_c^k(t):u_t^k}$.
 7: **end while**
**output** Node remaining in $G(t)$ as the root node $r_0$ of $G_0$.

---

By the definition of $K$-central nodes, as before we immediately have that under the uniform prior Algorithm 3 converges exponentially. We also have the following theorem, proven in Appendix F.

**Theorem 2.** *Let $G_0$ be the tree skeleton for the causal discovery problem. Consider a sequence of interventions determined by Algorithm 3. Define the running time (total number of interventions) of this Algorithm as $T_{CN}$ interventions. Let the running time of an optimal algorithm (that also performs $K$ interventions at each time) that finds $r_0$ be $T_{opt}$. Then*

$$T_{CN} \leq \lceil \log_{K+1} |G_0| \rceil \ \text{ and moreover, } \ \mathbb{E} T_{CN} \leq \frac{18}{7} \mathbb{E} T_{opt}.$$

*where the expectation is with respect to a uniform prior distribution $p_0(i) = 1/n$ over $i = 1, \dots, n$.*

### 3.3   Central node algorithm under node intervention restrictions

In many real-world applications, some subset $P$ of the nodes in $G_0$ cannot be intervened on (e.g. due to experimental limitations or cost). In Algorithm 5 (Appendix G), we extend the central node algorithm to this setting, modifying it to choose the best unrestricted node whenever the central node is restricted. We have the following theorem, proved in Appendix G.

**Theorem 3.** *Let $G_0$ be an undirected tree for the causal discovery problem. Let $P \subset G_0$ be a subset of nodes that are restricted from intervention. Assume that the probability that the root node is in $G_0 \setminus P$ (where $\setminus$ denotes set difference) is uniformly distributed. Define the running time (total number of interventions) of Algorithm 5 to be $T_{CN}$ interventions. Let the running time of an optimal (in terms of number of interventions) algorithm that finds $r_0$ be $T_{opt}$. Then*

$$\mathbb{E} T_{CN} \leq 3 \mathbb{E} T_{opt},$$

*where the expectation is with respect to a uniform prior distribution $p_0(i) = 1/n$ over $i = 1, \dots, n$.*

### 3.4   Central node algorithm for noisy observations

We now analyze the case in which an intervention on $X_i$ no longer gives noiseless information. This is a setting that arises in many applications. For simplicity, we restrict our discussion to the case in which the $X_i$ are binary variables, although our techniques may be applied to more general settings as well. Note that if an edge from one node to another is too weak, then no learning can occur. Hence we require the following condition on the noise:

**Condition 1** (Bounded edge strength). *We say that the edge strength of a tree $G$ is lower bounded by $\epsilon > 0$ if the following holds: for any nodes $i, j$ adjacent in the graph such that $i$ causes $j$, we have*

$$|\mathbb{P}(X_j = 1 \mid \mathsf{do}(X_i = 1)) - \mathbb{P}(X_j = 1)| > \epsilon.$$

Under the bounded edge strength condition, we have the following proposition indicating that repeating an intervention $\mathsf{do}(X_i = 1)$ a constant number of times is sufficient to good estimators of whether each branch around $X_i$ contains the root:

**Proposition 2.** *Under Condition 1, let $\delta_0 > 0$ be a desired soundness. Suppose $X_i$ has neighbors $u_1, \ldots, u_d \in N_G(X_i)$. Then with $\lceil 2 \log(1/\delta_0)/\epsilon^2 \rceil = O(\log(1/\delta_0)/\epsilon^2)$ samples from $\mathsf{do}(X_i = 1)$, we may output estimators $\hat{a}^{X_i:u_1}, \ldots, \hat{a}^{X_i:u_d} \in \{0, 1\}$ such that for each $j \in [d]$,*

$$\mathbb{P}(\hat{a}^{X_i:u_j} = 1 \mid R \in B_G^{X_i:u_j}) \geq 1 - \delta_0, \qquad \mathbb{P}(\hat{a}^{X_i:u_j} = 0 \mid R \notin B_G^{X_i:u_j}) \geq 1 - \delta_0,$$

*in other words the estimators $\hat{a}^{X_i:u_j}$ successfully identify whether the root $R$ is in branch $B_G^{X_i:u_j}$ or not with the desired soundness.*

Using this fact, we propose Algorithm 8 (in Appendix H) that slightly modifies the central node algorithm. For each intervention, we collect $\lceil 2 \log(1/\delta_0)/\epsilon^2 \rceil$ samples and update the posterior. We then add the following step: if the posterior probability of a node is close to 1, we flag it for inspection and temporarily remove it from consideration. Intuitively, this last step ensures that the current central node will never have too large a weight, so that for each intervention it is possible to lower-bound the total probability mass that is not contained in the branch containing the root, implying that each intervention always prunes away enough of the remaining probability mass. We then have the following theorem, proved in Appendix H.

**Theorem 4.** *Under Condition 1, Algorithm 8 takes $O(\log(n/\delta)/\epsilon^2)$ steps in expectation, and returns the true root node with probability at least $1 - \delta$.*

Note that if we were to directly apply the existing noisy graph search algorithms of Emamjomeh-Zadeh et al. (2016) to our model, then when applying Proposition 2 we would have to take our soundness parameter to be $\delta_0 \leq 1/\Delta$, where $\Delta$ is the maximum degree of the tree, and therefore our run-time guarantee would scale as $\log(\Delta) \log(n/\delta)/\epsilon^2$. Instead, our runtime does not depend on the degree of the tree. Furthermore, the rate is directly comparable to that found for the noiseless case (Theorem 1), up to the incorporation of the $\epsilon$ and $\delta$ parameters that control the uncertainty.

## 4 Empirical results

We consider several experimental settings and for each setting we simulate 200 random trees of $n$ nodes. Here we present a subset of the results, more are described in Appendix I. We generate an undirected tree with three different strategies: a) sampling uniformly from the space of undirected trees, b) generating power-law trees, and c) generating high degree $d = n/2$ random graphs and then creating an undirected version of the BFS tree. In this section we show only results for strategy a), but similar conclusions apply to the other strategies, given in Appendix I. Once we have a tree, we pick the root node uniformly at random. In this section we focus on binary random variables, where each variable is a function of its parent: if $X_{Pa_i} = 0$, then $X_i \sim \text{Bern}(\epsilon)$, else $X_i \sim \text{Bern}(1 - \epsilon)$, where for each variable we sample $\epsilon$ uniformly from $[\delta, 0.5 - \delta]$. The root node is distributed as $X_r \sim \text{Bern}(0.5)$. We show similar results with discrete variables in Appendix I.

Figure 3 shows the average number of interventions required to find the root node for three finite sample algorithms, all using our posterior update (Appendix B): a baseline algorithm that intervenes on a node randomly selected using the probability of being root in the current prior, the information greedy algorithm, implemented following the sampling strategy presented in Appendix C.2 with $N = 50$, and our central node algorithm presented in Algorithm 1. In the $n = 1000$ case, the information greedy algorithm was too computationally intensive and therefore omitted. Figure 4 shows the performance of the $K$-central node algorithm for varying $K$. Figure 5 shows the behavior of the finite sample extension of Algorithm 2 for $n = 50$ and different values of $\delta$, when we vary the number of interventional samples collected for each intervention from 1 (Algorithm 1) to 50. As expected, the behavior of the central node algorithm improves smoothly with the number of interventional samples, quickly converging to the performance of the noiseless Algorithm 2.

## 5 Conclusion

In this paper we proposed an active learning framework designed to reduce the number of interventions required to identify the directed tree specifying the causal model for a given set of variables. We presented algorithms for active learning when the observational data admits a tree model and proved that they can identify the causal graph with a number of interventions within a constant factor of the

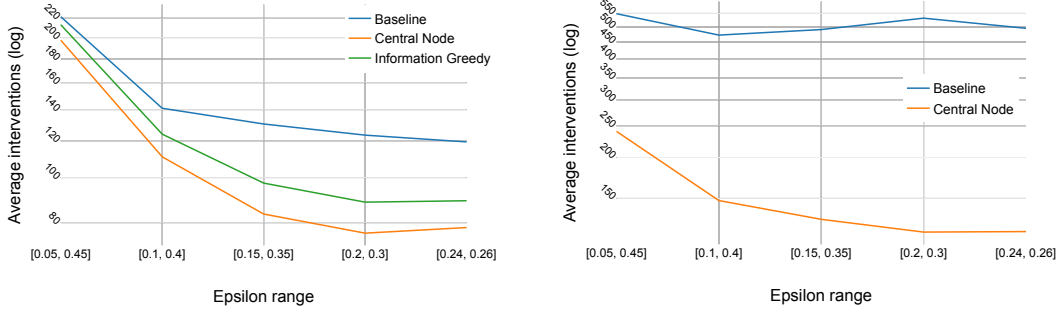

Figure 3: Average number of interventions for finite sample algorithms for varying ranges of $\epsilon$, $n = 30$ (left) and $n = 1000$ (right).

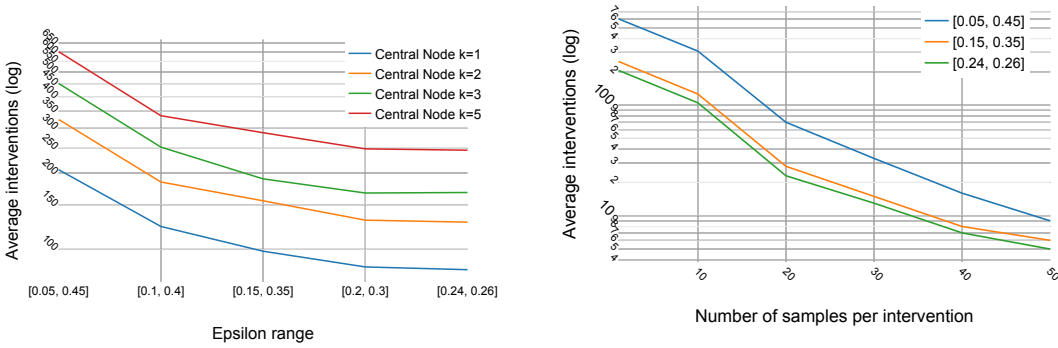

Figure 4: Average number of interventions for the K-central node algorithm for $K = 1, 2, 3, 5$ for $n = 50$ and varying ranges of $\epsilon$.

Figure 5: Finite sample extension of central node, varying number of interventional samples, $n = 50$, the curves represent different $\epsilon$ ranges.

(unknown) optimal procedure. As future work, we plan to extend our active learning approach and algorithms to more complex graph structures, such as chordal graphs. We also plan to extend our approach to the case where it is desirable to optimize the accuracy of regressions and/or predictions computed according to the model, instead of simply the correctness of the learned structure.

## Footnotes

[1] $\mathsf{do}(X = 1)$ is chosen for notational simplicity, as long as there is any value $a$ for which $\mathsf{do}(X = a)$ affects the effect variables, we can find it from the observational data and the theory will still hold.

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
