[Supplementary Material · SampleEfficientActiveLearningCausalTrees_supplement.pdf]

# Supplementary material for: Sample Efficient Active Learning of Causal Trees

## A  Algorithm for finding a central node

Algorithm 4 gives a simple way to find a central node in any given tree.

---

**Algorithm 4** Finding a central node

**input**  Undirected tree $G$ with some distribution $q$ over the nodes $i = 1, \ldots, n$.
1: Choose a node $v$ from $1, \ldots, n$. Find neighbors $N_G(v)$.
2: **while** $\max_{j \in N_G(v)} q(R \in B_G^{v:X_j}) \geq 1/2$ **do**
3:   $v \leftarrow \operatorname{argmax}_{j \in N_G(v)} q(R \in B_G^{v:X_j})$.
4: **end while**
**output**  Central node $v_c = v$.

---

## B  Proof of posterior update computation

After the intervention on $X_i$, the posterior update is given by the following:

$$
\begin{aligned}
P(R \in B_G^{X_i:Y} | \mathbf{X}, \mathsf{do}(X_i = 1)) &= \frac{P(R \in B_G^{X_i:Y} | \mathsf{do}(X_i = 1)) P(\mathbf{X} | R \in B_G^{X_i:Y}, \mathsf{do}(X_i = 1))}{P(\mathbf{X} | \mathsf{do}(X_i = 1))} \\
&= \frac{P(R \in B_G^{X_i:Y}) P(\mathbf{X} | R \in B_G^{X_i:Y}, \mathsf{do}(X_i = 1))}{P(\mathbf{X} | \mathsf{do}(X_i = 1))},
\end{aligned}
\tag{1}
$$

where the last step follows since the prior probability of the root node being in $B_G^{X_i:Y}$ is independent of the intervention.

Given the values of the neighbors $N_G(i)$ of $X_i$, the rest of the variables are independent of the value of $X_i$. Moreover, since $P(X_i = 1 | R \in B_G^{X_i:Y}, \mathsf{do}(X_i = 1)) = 1$ by definition, we can write:

$$
P(\mathbf{X} | R \in B_G^{X_i:Y}, \mathsf{do}(X_i = 1)) = P(N_G(i) | R \in B_G^{X_i:Y}, \mathsf{do}(X_i = 1)) P(\mathbf{X} \setminus \{N_G(i), X_i\} | N_G(i))
\tag{2}
$$

Since intervening at $X_i$ makes its parents independent of its children, if $R \in B_G^{X_i:Y}$ then $Y$ is independent of $N_G(i) \setminus Y$, giving

$$
\begin{aligned}
P(N_G(i) | R &\in B_G^{X_i:Y}, \mathsf{do}(X_i = 1)) \\
&= P(N_G(i) \setminus Y | R \in B_G^{X_i:Y}, \mathsf{do}(X_i = 1)) P(Y | R \in B_G^{X_i:Y}, \mathsf{do}(X_i = 1)) \\
&= P(N_G(i) \setminus Y | R \in B_G^{X_i:Y}, \mathsf{do}(X_i = 1)) P(Y).
\end{aligned}
\tag{3}
$$

The children of $X_i$ also become independent conditioned on $X_i$ giving

$$
P(N_G(i) \setminus Y | R \in B_G^{X_i:Y}, \mathsf{do}(X_i = 1)) = \prod_{A \in N_G(i) \setminus Y} P(A | X_i = 1).
\tag{4}
$$

Combining (1) through (4), for $Y \in N_G(i)$ the posterior update becomes

$$
\begin{aligned}
P(R \in B_G^{X_i:Y}|\mathbf{X}, \mathsf{do}(X_i = 1)) &= \frac{P(R \in B_G^{X_i:Y})P(\mathbf{X}|R \in B_G^{X_i:Y}, \mathsf{do}(X_i = 1))}{P(\mathbf{X}|\mathsf{do}(X_i = 1))} \\
&\propto P(R \in B_G^{X_i:Y})P(\mathbf{X}|R \in B_G^{X_i:Y}, \mathsf{do}(X_i = 1)) \\
&= P(R \in B_G^{X_i:Y})P(\mathbf{X} \setminus \{N_G(i), X_i\}|N_G(i))P(Y) \prod_{A \in N_G(i)\setminus Y} P(A|X_i = 1) \\
&\propto P(R \in B_G^{X_i:Y})P(Y) \prod_{A \in N_G(i)\setminus Y} P(A|X_i = 1) \\
&\propto P(R \in B_G^{X_i:Y})\frac{P(Y)}{P(Y|X_i = 1)},
\end{aligned}
\tag{5}
$$

where in the last step we have divided by $\prod_{A \in N_G(i)} P(A|X_i = 1)$ which is independent of $Y$.

For $Y$ being the intervention node $X_i$, all likelihoods of the neighbors $N_G(i)$ become conditional on $X_i = 1$ and the posterior update reduces to

$$
P(R = X_i|\mathbf{X}, \mathsf{do}(X_i = 1)) \propto P(R = X_i),
$$

where the constant of proportionality is the same as in (5).

Once normalized, we can distribute the update across the different nodes $Y$ in branch $B_G^{X_i:Y}$ according to their prior probabilities. $\square$

## C   Computation of expected information gain

The information greedy algorithm intervenes on the node that in expectation reduces the entropy of the posterior the most. Define the reduction in entropy as the difference between the entropy of the prior and the entropy of the posterior, i.e.

$$
\Delta H = H(P(R)) - H(P(R|\mathbf{X})),
$$

where $\mathbf{X}$ is the observed data. The goal of the information greedy approach is to choose the node that maximizes $\mathbb{E}\Delta H$, where the expectation is taken over both the true root and the observed data. In our experiments, we computed this expected information gain either via sampling or analytically.

### C.1   Analytical method

Since the expected change in entropy between a prior and posterior is equal to the mutual information between the observation and the quantity being estimated, we can write

$$
\mathbb{E}\Delta H = I(X; R|\mathsf{do}(X_i = 1)).
$$

Now, Section B showed that the only relevant information is contained in the set of neighbors $N_G(i)$ of $X_i$, letting us write

$$
\begin{aligned}
\mathbb{E}\Delta H =&I(N; R|\mathsf{do}(X_i = 1)) = H(N|\mathsf{do}(X_i = 1)) - H(N|R, \mathsf{do}(X_i = 1)) \\
=&H\left(\sum_{Y \in N_G(i)} P(R \in B_G^{X_i:Y})p(N|R \in B_G^{X_i:Y}, \mathsf{do}(X_i = 1))\right) \\
&- \sum_{Y \in N_G(i)} P(R \in B_G^{X_i:Y})H\left(p(N|R \in B_G^{X_i:Y}, \mathsf{do}(X_i = 1))\right)
\end{aligned}
$$

For certain models, e.g. binary graphs, the above expression may be simple to compute. For linear-Gaussian models, however, the first term becomes the entropy of a Gaussian mixture which does not have a closed form expression.

### C.2 Sampling method

For all $X_i \in \mathbf{X}$, simulate possible interventions on $X_i$ by repeating $j = 1 \ldots N$ times:

- Choose the candidate root branch $B_A$ using the current posterior $P(R \in B_A)$
- Simulate a fake interventional sample $P(\mathbf{X}|R \in B_A, \mathrm{do}(X_i = 1))$, assuming that the true root is in $B_A$. In practice, the only values that are useful are $N_G(i)$, the neighbours of $X_i$:
  - Sample $a$ from the observational distribution $P(A|R \in B_A, \mathrm{do}(X_i = 1)) = P(A)$.
  - For $Z_k \in N_G(i) \setminus A$, sample $z_k$ from $P(Z_k|R \in B_A, \mathrm{do}(X_i = 1)) = P(Z_k|X_i = 1)$.
- Simulate updating the posterior as in Lemma 1, for $Y \in N_G(i)$, including each $Z_k$ and $A$:
  - $P(R \in B_G^{X_i:Y}|a, z_k) \propto P(R \in B_G^{X_i:Y}) \frac{P(Y=y)}{P(Y=y|X_i=1)}$
- Compute entropy $H_{X_i,j}$ of the simulated posterior

Choose target as $X_t := \mathrm{argmax}_{X_i} \widehat{\mathbb{E}\Delta H_{X_i}} = \mathrm{argmax}_{X_i} \sum_{j=1}^{N} \Delta H_{X_i,j}$

## D   Proof of Proposition 1: Information greedy counterexample

The information greedy algorithm intervenes on the node that gives the most expected reduction in the entropy of the posterior distribution on the identity of the root node. For the first $K$ steps, this maximum information node is one of the $K$ $\ell_i$ nodes (Figure 2 in the main text). To see this, suppose that the algorithm is choosing a node on which to intervene after having intervened on nodes $\ell_1, \ldots, \ell_{i-1}$. Define $N_i$ to be the number of nodes remaining after the previous $i - 1$ interventions, observe that $(K + 2 - i)(\lceil e^{4K} \rceil + 1) \geq N_i \geq (K + 1 - i)\lceil e^{4K} \rceil$. Then the expected entropy reduction at node $\ell_i$ is

$$\frac{\lceil e^{4K} \rceil + 1}{N_i} \log N_i + \left( 1 - \frac{\lceil e^{4K} \rceil + 1}{N_i} \right) \left( \log N_i - \log(N_i - \lceil e^{4K} \rceil - 1) \right)$$

$$= \log N_i - \left( 1 - \frac{\lceil e^{4K} \rceil + 1}{N_i} \right) \log(N_i - \lceil e^{4K} \rceil - 1)$$

$$\geq \frac{1}{K + 2 - i} \log \left( (K + 1 - i)\lceil e^{4K} \rceil \right) \geq \frac{4K}{K + 1} \geq 2,$$

since an intervention on $\ell_i$ finds the root with probability $\frac{\lceil e^{4K} \rceil + 1}{N_i}$, and the entropy of a uniform distribution on $N_i$ nodes is $\log N_i$. Compare this to the expected entropy reduction that is bounded above by $\log 4$ at each other node (since the maximum edge degree is 3 at all nodes other than the $K$ $\ell_i$ nodes). Therefore, the information greedy algorithm will first intervene at one of the $\ell_i$. If the intervention does not immediately discover the root, it will indicate that the root is towards the center of the graph. The algorithm will then choose another of the $\ell_i$ since these remain the nodes with largest expected information gain. This process repeats until either the root is found or all the $\ell_i$ have been intervened on. Hence, in expectation at least $K/2$ steps are needed. □

## E   Proof of Theorem 1: Central node algorithm in the noiseless setting

First, we prove the exponential convergence of the central node algorithm. By the definition of the central node, each branch has probability mass no greater than $\frac{1}{2}$. Since the noiseless interventions eliminate all branches except one (if the intervention happens to be on the root node, then all branches are eliminated and the algorithm stops), each intervention must eliminate all but $\frac{1}{2}$ of the remaining probability mass. For a starting prior uniform on $|G_0|$ nodes,

$$|G(t)| \leq \frac{|G_0|}{2^t},$$

and the bound results.

We now relate $T_{CN}$ to $T_{opt}$. From Algorithm 2, let $t$ be the number of interventions the central node algorithm has taken at the current time.

Recall from Algorithm 2 that $G(t)$ is the tree of possible source nodes remaining in $G_0$ after $t$ interventions have been taken. Choose an optimal algorithm, and let $G_{opt}(\tau)$ be the tree of possible source nodes remaining in $G_0$ after the optimal algorithm has taken $\tau$ interventions. The $t - th$ node Algorithm 2 intervenes on is the central node $v_c(t)$, similarly let the $\tau - th$ node the optimal algorithm intervenes on be denoted $v_o(\tau)$.

Define $\tau(t)$ as the largest number of interventions the optimal algorithm takes while still preserving the inequality

$$G_{opt}(\tau(t)) \supseteq G(t).$$

In other words, $\tau(t)$ is such that Algorithm 2 has determined the root node $r_0$ lies in a set $G(t)$ contained in the set of nodes that the optimal algorithm deems possible after $\tau(t)$ interventions. We proceed inductively, bounding the increase in $\tau$ for each increase in $t$.

Begin by noting that $G_{opt}(\tau(0)) = G(0)$, hence $\tau(0) = 0$. At time $t$, we have $G_{opt}(\tau(t)) \supseteq G(t)$. Algorithm 2 then intervenes on the central node $v_c(t+1)$ of $G(t)$, observing $u_{t+1}$ and forming $G(t+1) = B_{G(t)}^{v_c(t+1):u_{t+1}}$. Meanwhile, the optimal algorithm intervenes on some node $v_o(\tau(t) + 1)$, yielding $G_{opt}(\tau(t) + 1)$.

If $v_o(\tau(t) + 1) \notin G(t+1)$, then $G_{opt}(\tau(t) + 1) \supset G(t+1)$ since any vertex the optimal algorithm eliminated as a possible root node at step $\tau(t+1)$ must lie on a branch from $v_o(\tau(t+1))$ that does not include $v_c(t+1)$, otherwise every node in $G(t+1)$ would be eliminated which would be a contradiction. Hence an intervention on $v_o(\tau(t)+1)$ cannot eliminate *any* nodes in $G(t+1)$ as possible root nodes. We then have that $\tau(t+1) = \tau(t)+1$ maintains the inequality $G_{opt}(\tau(t+1)) \supseteq G(t+1)$.

If $v_o(\tau(t) + 1) \in G(t+1)$, we can simply set $\tau(t+1) = \tau(t)$ to have the optimal algorithm wait an additional step, since $G_{opt}(\tau(t+1)) = G_{opt}(\tau(t)) \supseteq G(t) \supset G(t+1)$.

Using induction, we can now bound $\tau(t)$ in expectation. Since $v_c(t+1)$ is a central node of $G(t)$, if $v_c(t+1) \neq v_o(t+1)$ the probability that the central node algorithm eliminates the branch containing the node $v_o(t+1)$ the optimal algorithm intervened on is $\geq 1/2$. Hence $P(\tau(t+1)-\tau(t) = 1) \geq 1/2$, where the probability is based on the root node $r_0$ having a uniform prior distribution $p_0(i) = 1/n$. Hence $\mathbb{E}(\tau(t)) \geq t/2$ for all $t$, yielding $\mathbb{E}T_{opt} > \frac{1}{2}\mathbb{E}T_{CN}$ since both algorithms terminate when $G(t) = G(\tau(t)) = \{r_0\}$. $\qquad\square$

# F  Proof of Theorem 2: $K$-central node algorithm in the noiseless setting

From Algorithm 3, let $t$ be the number of steps the central node algorithm has taken at the current time.

Recall from Algorithm 3 that $G(t)$ is the tree of possible source nodes remaining in $G_0$ after $t$ interventions have been taken. Choose an optimal algorithm, and let $G_{opt}(\tau)$ be the tree of possible source nodes remaining in $G_0$ after the optimal algorithm has taken $\tau$ interventions. The $t - th$ set of nodes Algorithm 3 intervenes on is the set of $K$-central nodes $v_c^k(t)$, similarly let the $\tau - th$ set of nodes the optimal algorithm intervenes on be denoted $v_o^k(\tau)$.

In order to proceed with the proof, as a baseline we allow the central node algorithm to take *two* steps for every step taken by the optimal algorithm, and then will slow the optimal algorithm down by a factor $\tau(t)$ such that the slowed optimal algorithm is strictly worse than the central algorithm. Hence, define $\tau(t)$ as the largest number of interventions the optimal algorithm takes while still preserving the inequality

$$G_{opt}(\tau(t)) \supseteq G(2t).$$

In other words, $\tau(t)$ is such that Algorithm 3 has determined the root node $r_0$ lies in a set $G(2t)$ contained in the set of nodes that the optimal algorithm deems possible after $\tau(t)$ interventions. We proceed inductively, bounding the increase in $\tau$ for each increase in $t$.

Begin by noting that $G_{opt}(\tau(0)) = G(0)$, hence $\tau(0) = 0$. At time $t$, we have $G_{opt}(\tau(t)) \supseteq G(2t)$. The first step of Algorithm 3 then intervenes on the $K$-central nodes $v_c^k(2t + 1)$ of $G(2t)$, observing $u_{2t+1}^k$ and forming $G(2t + 1)$. The second step intervenes on the $K$-central nodes $v_c^k(2t + 2)$ of $G(2t + 1)$, observing $u_{2t+2}^k$ and forming $G(2(t + 1))$. Meanwhile, the optimal algorithm intervenes on some $K$ nodes $v_o^k(\tau(t) + 1)$, yielding $G_{opt}(\tau(t) + 1)$. Note that after the two steps the central

node algorithm takes, by the definition of $K$-central nodes, the mass of $G(2(t+1))$ is at most $\frac{1}{(K+1)^2}$ that of $G(2t)$.

If none of the $v_o^k(\tau(t)+1)$ are in $G(2(t+1))$ or if the central node algorithm identifies the true node, then by necessity (as in the previous central node proof) $G_{opt}(\tau(t)+1) \supseteq G(2(t+1))$. We then have that $\tau(t+1) = \tau(t) + 1$ maintains the inequality $G_{opt}(\tau(t+1)) \supseteq G(2(t+1))$.

If at least one of the $v_o^k(\tau(t)+1) \in G(2(t+1))$, we can simply set $\tau(t+1) = \tau(t)$ to have the optimal algorithm wait an additional step, since $G_{opt}(\tau(t+1)) = G_{opt}(\tau(t)) \supseteq G(2t) \supset G(2(t+1))$.

Using induction, we can now bound $\tau(t)$ in expectation. Since the $v_c^k(2(t+1))$ and $v_c^k(2(t+2))$ are $K$-central nodes of $G(2t)$ and $G(2t+1)$ respectively, each potential $G(2(t+1)$ (excluding cases where one of the intervened nodes has been the root, in which case the algorithm would terminate) has at most probability mass $\frac{1}{(K+1)^2}$. The $K$ nodes chosen by the optimal algorithm can be contained in at most $K$ of these with total mass at most $\frac{K}{(K+1)^2}$. Hence, the probability that either the central node algorithm terminates or none of the $v_o^k(\tau(t)+1)$ are in $G(2(t+1))$ is at least $1 - \frac{K}{(K+1)^2} \geq 7/9$. Hence $P(\tau(t+1) - \tau(t) = 1) \geq 7/9$, where the probability is based on the root node $r_0$ having a uniform prior distribution $p_0(i) = 1/n$. Hence $\mathbb{E}(\tau(t)) \geq \frac{7}{9}t$ for all $t$, yielding $\mathbb{E}T_{opt} > \frac{7}{18}\mathbb{E}T_{CN}$ since both algorithms terminate when $G(t) = G(\tau(t)) = \{r_0\}$ and we have slowed down the central node algorithm by a factor of two. $\qquad\square$

# G  Proof of Theorem 3: Central node algorithm in the noiseless setting with restricted nodes

---
**Algorithm 5** Node-Restricted Central Node Algorithm
---
**input** Observational tree skeleton $G_0$. Restricted set of nodes $P \subset V$
1:   $t \leftarrow 0$, $G(0) \leftarrow G_0$.
2: **while** $G(t)$ contains more than one node **do**
3:     $t \leftarrow t + 1$.
4:     Find a central node $v_c(t)$ of $G(t-1)$ under the uniform distribution over unrestricted nodes (Algorithm 4).
5:     **if** $v_c(t) \notin P$ **then**
6:       Intervene on $v_c(t)$ and observe direction of root node $u_t \in \{v_c(t)\} \cup N_{G(t-1)}(v_c(t))$.
7:       Set $G(t) \leftarrow B_{G(t-1)}^{v_c(t):u_t}$.
8:     **else**
9:       Form the tree $G'$ as $G(t-1)$ rooted at $v_c(t)$. Now, if we remove the connected component in $P \cap G'$ containing $v_c(t)$, we get $\ell(t)$ possible rooted subtrees of $G'$, let us call them $T_1, T_2 \ldots T_{\ell(t)}$. Let the roots (induced by $v_c(t)$) of each of these trees be $r_1, r_2 \ldots r_{\ell(t)}$. Without loss of generality, suppose that the probability mass of each of these subtrees with respect to the posterior distribution of the root node at time $t-1$ be $p_1 \geq p_2 \geq p_{\ell(t)}$.
10:      Sequentially intervene on the roots $r_1, r_2 \ldots r_{\ell(t)}$ in order until the subtree containing the root is found. Suppose it takes $s(t)$ steps. Let $\tilde{r}$ be the root of the tree that has the root node.
11:      $G(t+s(t)-1) \leftarrow B_{G(t+s(t)-2)}^{\tilde{r}:u_{t+s(t)-1}}$.
12:     **end if**
13: **end while**
**output** Node remaining in $G(t)$ as the root node $r_0$ of $G_0$.
---

From Algorithm 5, let $t$ be the number of interventions the central node algorithm has taken at the current time.

Recall from Algorithm 5 that $G(t)$ is the tree of possible source nodes remaining in $G_0$ after $t$ interventions have been taken. Choose an optimal algorithm, and let $G_{opt}(\tau)$ be the tree of possible source nodes remaining in $G_0$ after the optimal algorithm has taken $\tau$ interventions. The $t-th$ node Algorithm 5 intervenes on is the central node $v_c(t)$, similarly let the $\tau-th$ node the optimal algorithm intervenes on be denoted $v_o(\tau)$.

Define $\tau(t)$ as the largest number of interventions the optimal algorithm takes while still preserving the inequality

$$G_{opt}(\tau(t)) \supseteq G(t).$$

We will prove the invariant: $\mathbb{E}[\tau(t)] \geq t/3$ inductively.

**Base Case:** Begin by noting that $G_{opt}(\tau(0)) = G(0)$, hence $\tau(0) = 0$. At time $t$, we have $G_{opt}(\tau(t)) \supseteq G(t)$. Now, we have the following cases:

**Inductive Step:** Suppose, the hypothesis is true for $t$. We show it is true for $t + 1$.

**Case 1:** Algorithm 5 intervenes on the central node $v_c(t+1)$ of $G(t)$, observing $u_{t+1}$ and forming $G(t+1) = B_{G(t)}^{v_c(t+1):u_{t+1}}$. Meanwhile, the optimal algorithm intervenes on some node $v_o(\tau(t)+1)$, yielding $G_{opt}(\tau(t) + 1)$.

If $v_o(\tau(t)+1) \notin G(t+1)$, then $G_{opt}(\tau(t)+1) \supset G(t+1)$ since any vertex the optimal algorithm eliminated as a possible root node at step $\tau(t+1)$ must lie on a branch from $v_o(\tau(t+1))$ that does not include $v_c(t+1)$, otherwise every node in $G(t+1)$ would be eliminated which would be a contradiction. Hence an intervention on $v_o(\tau(t)+1)$ cannot eliminate *any* nodes in $G(t+1)$ as possible root nodes. We then have that $\tau(t+1) = \tau(t) + 1$. This maintains the invariance $G_{opt}(\tau(t+1)) \supseteq G(t+1)$.

If $v_o(\tau(t)+1) \in G(t+1)$, we can simply set $\tau(t+1) = \tau(t)$ to have the optimal algorithm wait an additional step, since $G_{opt}(\tau(t+1)) = G_{opt}(\tau(t)) \supseteq G(t) \supset G(t+1)$.

Since $v_c(t+1)$ is a central node of $G(t)$, if $v_c(t+1) \neq v_o(t+1)$ the probability that the central node algorithm eliminates the branch containing the node $v_o(t+1)$ the optimal algorithm intervened on is $\geq 1/2$. Hence $P(\tau(t+1) - \tau(t) = 1) \geq 1/2$, where the probability is based on the root node $r_0$ having a uniform prior distribution $p_0(i) = 1/n$. Therefore, $\mathbb{E}[\tau(t+1)] \geq \mathbb{E}[\tau(t)] + \frac{1}{2} \geq t/3 + 1/2 \geq \frac{t+1}{3}$

**Case 2:** At step $t+1$, $v_c(t+1)$ is restricted in $G_t$. Let us consider the tree $G_t$ rooted at $v_c(t+1)$. Now, If we remove the nodes in $P \cap G_t$, we get $\ell(t+1)$ possible rooted subtrees, let us call them $T_1, T_2 \ldots T_{\ell(t+1)}$. Let the root of each of these trees be $r_1, r_2 \ldots r_{\ell(t+1)}$. Without loss of generality, suppose that the probability mass of each of these subtrees with respect to the posterior distribution of the root node at time $t$ be $p_1 \geq p_2 \geq p_{\ell(t+1)}$. Our Central node algorithm intervenes sequentially on the roots $r_1, r_2 \ldots r_{\ell(t+1)}$ in that order until it finds the tree that contains the root node. Let $s(t+1)$ be the number of interventions after time $t$ that the central node algorithm takes before it finds the tree with the root node. Let $s(\tau(t))$ be the number of interventions before (but after $\tau(t)$) the first intervention is made on the tree with the root node by the optimal algorithm

Clearly, $G(t + s(t+1) + 1) \subset G_{\text{opt}}(\tau(t) + s(\tau(t)))$.

We have:

$$\tau(t + k + 1) \leq \tau(t) + s(\tau(t)), \ \forall k < s(t+1).$$
$$\tau(t + s(t+1) + 1) = \tau(t) + s(\tau(t)) \tag{6}$$

Clearly, $G(t+k+1) \subset G(t+s(t+1)+1) \subset G_{\text{opt}}(\tau(t)+s(\tau(t))) \subset G_{\text{opt}}(\tau(t+k+1))$

Because the central node algorithm intervenes in the order of decreasing probability of root being found in the trees $T_1, T_2 \ldots T_{\ell(t+1)}$, we have : $\mathbb{E}[s(\tau(t))|\mathcal{F}_t] \geq \mathbb{E}[s(t+1)|\mathcal{F}_t] = (p_1 + 2p_2 \ldots) - 1 \geq 1 - p_1 \geq \frac{1}{2}$. This implies $3\mathbb{E}[s(\tau(t))|\mathcal{F}_t] \geq \mathbb{E}[s(t+1)|\mathcal{F}_t] + 1$

This means that: $\mathbb{E}[\tau(t) + s(\tau(t))] \geq \frac{1}{3}\mathbb{E}[t + s(t+1) + 1]$. Therefore, the following holds:

$$\mathbb{E}[\tau(t+k+1)] \stackrel{a}{=} \mathbb{E}[\tau(t+s(t+1)+1)] = \mathbb{E}[\tau(t)+s(\tau(t))] \geq \frac{1}{3}\mathbb{E}[t+s(t+1)+1]$$

$$\geq \frac{1}{3}\mathbb{E}[\tau(t+k+1)], \ 0 \leq k \leq s(\tau(t)) \tag{7}$$

(a) - This is from (6).

This proves the induction step. The invariance implies the final result. $\qquad \square$

# H Central node algorithm in the noisy setting

In this section, we show how to modify the central node algorithm so as to obtain error and runtime guarantees in the noisy setting with a binary alphabet. First, recall the condition on the noise that we require:

**Condition 2** (Bounded edge strength, restatement of Condition 1). *We say that a tree $G$'s edge strength is lower bounded by $\epsilon > 0$ if the following holds: for any nodes $i, j$ adjacent in the graph such that $i$ causes $j$, we have*

$$|\mathbb{P}(X_j = 1 \mid \mathsf{do}(X_i = 1)) - \mathbb{P}(X_j = 1)| > \epsilon.$$

Under the bounded edge strength condition, given interventional data at a node $v$ we may efficiently construct good estimators of whether each branch $B_G^{v:u}$ contains the root:

**Proposition 3** (Restatement of Proposition 2). *Under Condition 1, let $\delta_0 > 0$ be a desired soundness. Suppose $X_i$ has neighbors $u_1, \ldots, u_d \in N_G(X_i)$. Then with $\lceil 2\log(1/\delta_0)/\epsilon^2 \rceil = O(\log(1/\delta_0)/\epsilon^2)$ samples from $\mathsf{do}(X_i = 1)$, we may output estimators $\hat{a}^{X_i:u_1}, \ldots, \hat{a}^{X_i:u_d} \in \{0, 1\}$ such that for each $j \in [d]$,*

$$\mathbb{P}(\hat{a}^{X_i:u_j} = 1 \mid R \in B_G^{X_i:u_j}) \geq 1 - \delta_0, \qquad \mathbb{P}(\hat{a}^{X_i:u_j} = 0 \mid R \notin B_G^{X_i:u_j}) \geq 1 - \delta_0.$$

*Proof.* For each $j \in [d]$, let $x_{u_j,1}, \ldots, x_{u_j,K} \in \{0, 1\}$ be the observations of $X_{u_j}$ on the $K$ different interventions. Write

$$p = \mathbb{P}[X_{u_j} = 1 \mid \mathsf{do}(X_i = 1), R \in B_G^{X_i:u_j}], \quad q = \mathbb{P}[X_{u_j} = 1 \mid \mathsf{do}(X_i = 1), R \notin B_G^{X_i:u_j}],$$

and by the bounded noise condition assume without loss of generality that $p - q \geq \epsilon$.

Let $s_{u_j} = \sum_{j=k}^{K} x_{u_j,k}$, and let $\hat{a}^{X_i:u_j} = 1(s_{u_i} \geq (q + \epsilon/2)K)$. By Hoeffding bounds,

$$\mathbb{P}[\hat{a}^{X_i:u_j} = 0 \mid R \in B_G^{X_i:u_j}] \leq \exp(-\epsilon^2 K/2), \quad \mathbb{P}[\hat{a}^{X_i:u_j} = 1 \mid R \notin B_G^{X_i:u_j}] \leq \exp(-\epsilon^2 K/2).$$

$\square$

We are now ready to present our algorithm, which is inspired by the algorithms of Ben-Or & Hassidim (2008) and Emamjomeh-Zadeh et al. (2016) for noisy search in graphs. These papers' noise models differ from ours: in their case each noisy node query tells the algorithm whether it is the root node, or, if not, on which branch the root node lies. The algorithm of Emamjomeh-Zadeh et al. (2016) runs in time $O((\log(n) + \log(1/\delta)^2)/(p - 1/2)^2)$, where $p > 1/2$ is the probability that a noisy query is correct and $\delta$ is the desired bound on the error probability of the algorithm. Directly applying this algorithm to our setting, we would need to perform $O((\log \Delta)/\epsilon^2)$ interventions in order to simulate one noisy query, where $\Delta$ is the maximum degree of the tree. This would give a total run-time of $O((\log \Delta)(\log(n) + \log(1/\delta)^2)/\epsilon^2)$. In contrast, the algorithm that we present below runs in time $O(\log(n/\delta)/\epsilon^2)$, which avoids the sub-optimal dependence on $\delta$, and does not depend on the maximum degree of the tree.

We now present Algorithm 6, which is the multiplicative weights algorithm at the core of the noisy central node algorithm. In Algorithm 6, we maintain a weight $q_t(w)$ for each node $w$ and iteration $t$. On each iteration $t$, we find a central node $v_c(t)$ of $q_{t-1}(\cdot)$, so that each of the branches around $v_c(t)$ has at most half of the total weight. If the weight of $v_c(t)$ is large, we temporarily remove $v_c(t)$ from consideration and flag it for later. Otherwise, we intervene on $v_c(t)$, compute estimators $\hat{a}^{v_c(t):u_i}$ as in Proposition 2 for each of the branches around $v_c(t)$, and lower the weight of the branches $B_G^{v_c(t):u_i}$ for which $\hat{a}^{v_c(t):u_i} = 0$.

Our error guarantees for the multiplicative weights algorithm will follow from a potential argument. We will prove that the sum of the weights $\Psi_T := \sum_{w \in S} q_T(w)$ at the end of the multiplicative weights algorithm is usually very small, but that $q_T(R)$ is usually very large if $R$ is not added to the output set $M$. Since $q_T(R) \leq \Psi_T$ we conclude that $R$ is in the output set most of the time:

**Proposition 4.** *Suppose that we run the multiplicative weights algorithm (Algorithm 6) for $T$ iterations and with input set $S$ and parameters $\tau, \delta_0, \eta \in [0, 1]$. For all $t \leq T$ define the potential function $\Psi_t := \sum_{w \in S} q_t(w)$. Then*

$$\mathbb{E}[\Psi_T] \leq \gamma^T,$$

*where $\gamma = \gamma(\tau, \delta_0, \eta) := \max(1 - \tau, 1 - (1/2 - \tau) \cdot (1 - \delta_0) \cdot (1 - \eta))$.*

**Algorithm 6** MULTIPLICATIVE-WEIGHTS algorithm

---

**input** Observational tree $G$ for a model whose noise is bounded by $\epsilon > 0$ (Condition 1). Set
$S \subset V(G)$. Number of iterations $T$. Parameters $\tau, \delta_0, \eta \in [0,1]$.
1: $M \leftarrow \emptyset$.
2: $q_0(w) \leftarrow 1/|S|, \forall w \in S$.
3: **for** $1 \le t \le T$ **do**
4:     Identify central node index $v_c(t)$ of $G$ with respect to $q_{t-1}$ (Algorithm 4).
5:     **if** $q_{t-1}(v_c(t)) \ge \tau \cdot (\sum_{w \in S} q_{t-1}(w))$ **then**
6:         $M \leftarrow M \cup \{v_c(t)\}$.
7:         $q_t(v_c(t)) \leftarrow 0$.
8:         $q_t(w) \leftarrow q_{t-1}(w), \forall w \ne v_c(t)$.
9:     **else**
10:        $q_t(v_c(t)) \leftarrow q_{t-1}(v_c(t))$.
11:        Intervene $\lceil 2\log(1/\delta_0)/\epsilon^2 \rceil$ times with $\mathsf{do}(v_c(t) = 1)$, and for all $i \in [d]$ obtain an estimator
           $\hat{a}^{v_c(t):u_i} \in \{0,1\}$ for $1(R \in B_G^{v_c(t):u_i})$, with soundness parameter $\delta_0$ as in Proposition 2.
12:        **for** $1 \le i \le d$ and $w \in B_G^{v_c(t):u_i}$ **do**
13:            **if** $\hat{a}^{v_c(t):u_i} = 0$. **then**
14:               $q_t(w) \leftarrow \eta \cdot q_{t-1}(w)$.
15:            **else**
16:               $q_t(w) \leftarrow q_{t-1}(w)$.
17:            **end if**
18:        **end for**
19:     **end if**
20: **end for**
**output** $M$.

---

*Proof.* Let $\mathcal{F}_t$ be the filtration corresponding to the state of the algorithm at the end of the $t$th step. For any $t \ge 1$, consider $\mathbb{E}[\Psi_t \mid \mathcal{F}_{t-1}]$. Let $E_t$ denote the event that $q_{t-1}(v_c(t)) > \tau \cdot \Psi_{t-1}$. If $E_t$ holds then $v_c(t)$ is removed so

$$\mathbb{E}[\Psi_t \mid \mathcal{F}_{t-1}, E_t] \le (1 - \tau) \cdot \Psi_{t-1}. \tag{8}$$

Otherwise, for each node $w \in S$, let $D_{t,w}$ be the event that, on iteration $t$, the node $w$ is in a branch $B_G^{v_c(t):u_i}$ that does not contain the root. In this case, with probability $\ge 1 - \delta_0$, the estimator $\hat{a}^{v_c(t):u_i} = 0$, in which case the weight of $w$ is decreased by a factor of $\eta$:

$$\mathbb{E}[q_t(w) \mid \mathcal{F}_{t-1}, \neg E_t, D_{t,w}] \le (1 - (1 - \delta_0) \cdot (1 - \eta)) \cdot q_{t-1}(w).$$

Since $v_c(t)$ is a central node of weight $q_{t-1}(v_c(t)) \le \tau \cdot \Psi_{t-1}$ and no branch has weight more than $\Psi_{t-1}/2$ because $v_c(t)$ is the central node, the total weight of the branches not containing the root node is $\ge (\frac{1}{2} - \tau) \cdot \Psi_{t-1}$. Therefore by linearity of expectation

$$\mathbb{E}[\Psi_t \mid \mathcal{F}_{t-1}, \neg E_t] \le (1 - (1/2 - \tau) \cdot (1 - \delta_0) \cdot (1 - \eta)) \cdot \Psi_{t-1}. \tag{9}$$

Overall, (8) and (9) allow us to conclude that

$$\mathbb{E}[\Psi_t | \mathcal{F}_{t-1}] \le \max(1 - \tau, (1 - (1/2 - \tau) \cdot (1 - \delta_0) \cdot (1 - \eta))) \cdot \Psi_{t-1} = \gamma \Psi_{t-1},$$

and so $\mathbb{E}[\Psi_T] = \mathbb{E}[\mathbb{E}[\Psi_T \mid \mathcal{F}_{t-1}]] \le \gamma \mathbb{E}[\Psi_{t-1}] \le \cdots \le \gamma^T \mathbb{E}[\Psi_0] = \gamma^T$.

$\square$

**Proposition 5.** *Suppose that we run the multiplicative weights algorithm (Algorithm 6) for $T$ iterations and with input set $S \ni R$ and parameters $\tau, \delta_0, \eta \in [0,1]$. Let $M$ be the set outputted by the algorithm. Then for any $\lambda \ge 0$,*

$$\mathbb{P}[q_T(R) \le \eta^{(\delta_0 + \lambda)T}/|S| \mid R \notin M] \le \exp(-2T\lambda^2)$$

*Proof.* Since $R \notin M$ holds, the weight of $R$ is only ever lowered by the multiplicative weight update. On any given step, the probability that $q_{t-1}(R)$ is lowered is at most $\delta_0$, by Proposition 2. Let $\xi$ be the number of steps on which the weight of $R$ is decreased. By a Hoeffding bound,

$$\mathbb{P}[\xi \ge (\delta_0 + \lambda)T \mid R \notin M] \le \exp(-2T\lambda^2).$$

The theorem follows from $q_T(R) = \eta^\xi \cdot q_0(R) = \eta^\xi/|S|$.

$\square$

**Proposition 6.** *Suppose that we run the multiplicative weights algorithm (Algorithm 6) for $T$ iterations and with input set $S \ni R$ and parameters $\tau, \delta_0, \eta \in [0,1]$. Let $\lambda > 0$ such that $\eta^{(\delta_0+\lambda)} > \gamma$ and*

$$T \geq \max\left( \frac{\log(20|S|)}{\log(\eta^{(\delta_0+\lambda)}/\gamma)}, \frac{\log 2}{2\lambda^2} \right).$$

*Then if $M$ is the set outputted by the algorithm,*

$$\mathbb{P}[R \notin M] \leq 1/10.$$

*Proof.* Let $A := \eta^{(\delta_0+\lambda)T}/|S|$. By Proposition 4 and a Markov bound,

$$\mathbb{P}[\Psi_T \geq A] \leq \gamma^T/A \leq 1/20.$$

Since $0 \leq q_T(R) \leq \psi_T$, we have $\mathbb{P}[q_T(R) \geq A] \leq 1/8$. Therefore, by Proposition 5 and Bayes' rule

$$\mathbb{P}[R \notin M] \leq \frac{1}{8 \cdot \mathbb{P}[q_T(R) \geq A \mid R \notin M]} \leq \frac{1}{20 \cdot (1 - 1/2)} \leq 1/10.$$

$\square$

For the sake of concreteness, let us fix values $\tau, \delta_0, \eta$, and $\lambda$. First, fix $\tau = \tau(\delta_0, \eta) = \frac{1}{2} \cdot \left( \frac{(1-\delta_0)\cdot(1-\eta)}{1+(1-\delta_0)\cdot(1-\eta)} \right)$ because this minimizes $\gamma(p_0, \delta, \tau)$. Then choose, say, $\eta = 3/4$, $\delta_0 = 1/10$, and $\lambda = 1/10$. One may verify that $\eta^{\delta_0+\lambda} > 0.94 > 0.91 > \gamma$. In particular, if we choose these parameters then when $T \geq 60\log(20|S|)$ we have $\mathbb{P}[R \notin M] \leq 1/10$.

We now apply the multiplicative weights algorithm in Algorithm 7, which returns the root with probability $\geq 2/3$.

---

**Algorithm 7** Noisy Central Node Algorithm. Finds root with probability $\geq 2/3$.

**input** Observational tree $G$.
1: Choose parameters $\delta_0 = 1/10, \eta = 3/4$ and $\tau = \left( \frac{(1-\delta_0)\cdot(1-\eta)}{1+(1-\delta_0)\cdot(1-\eta)} \right)$.
2: $M_1 \leftarrow$ MULTIPLICATIVE-WEIGHTS$(S = V(G), T = 60\log(20n), \tau, \delta_0, \eta)$.
3: $M_2 \leftarrow$ MULTIPLICATIVE-WEIGHTS$(S = M_1, T = 60\log(20|M_1|), \tau, \delta_0, \eta)$.
4: Let $\delta_1 \leftarrow 1/(10|M_2|^2)$.
5: **for** $v \in M_2$ **do**
6:     Let $u_1, \dots, u_d$ be the neighbors of $v$.
7:     Do $\lceil 2\log(1/\delta_1)/\epsilon^2 \rceil$ interventions $do(v = 1)$ to get estimators $\hat{a}^{v:u_i}$ of $1(R \in B_G^{v:u_i})$ with soundness parameter $\delta_1$, as in Proposition 2.
8:     **if** $\hat{a}^{v:u_i} = 0$ for all $u_i$ such that $M_2 \cap B_G^{v:u_i} \neq \emptyset$ **then**
9:         output $v$.
10:     **end if**
11: **end for**
**output** "not found"

---

**Proposition 7.** *Algorithm 7 returns $R$ with probability $\geq 2/3$, and uses $O((\log n)/\epsilon^2)$ queries.*

*Proof.* **Correctness** Let $E$ be the event that $R \in M_2$.

Consider the set $A := \{(v, u_i) : v \in M_2, B_G^{v:u_i} \cap M_2 \neq \emptyset\}$. Let $E'$ be the event that

$$\hat{a}^{v:u_i} = 1(R \in B_G^{v:u_i}) \quad \forall (v, u_i) \in A.$$

Suppose that $E$ and $E'$ both hold. Then the output $\hat{R}$ of the algorithm is equal to the true root $R$. To see this, consider the loop in steps 5-11.

- If $v = R$, then since $E'$ holds we will have $\hat{a}^{v:u_i} = 0$ for all neighbors $u_i$ of $v$ such that $M_2 \cap B_G^{v:u_i} \neq \emptyset$. Therefore, the algorithm outputs $v$ in step 9.

- If $v \neq R$, then let $u_i$ be the neighbor of $v$ such that $R \in B_G^{v:u_i}$. Since $R \in M_2$ by event $E$, we have $B_G^{v:u_i} \cap M_2 \neq \emptyset$, and also $\hat{a}^{v:u_i} = 1$ by event $E'$. Thus, the algorithm does not output $v$ in step 9.

So it suffices to show that $\mathbb{P}[E \cap E'] \geq 2/3$.

By two applications of Proposition 6 and a union bound, $\mathbb{P}[E] \geq 8/10$.

We have chosen $\delta_1$ so that for any $(v, u_i) \in A$, $\mathbb{P}[\hat{a}^{v:u_i} \neq 1(R \in B_G^{v:u_i})] \leq \delta_1 = 1/|M_2|^2$. Note that $|A| \leq |M_2|^2$, since for each node $v \in M_2$ at most $|M_2|$ of the branches $B_G^{v:u_i}$ contain a node in $M_2$. Therefore, by a union bound $\mathbb{P}[E'] \geq 9/10$.

By a union bound, $\mathbb{P}[E \cap E'] \geq 2/3$.

**Query complexity** Note that $|M_2| = O(\log \log n)$ since $|M_1| = O(\log n)$ and $|M_2| = O(\log |M_1|)$, because at most one vertex is added to the output set on each iteration of the multiplicative weights algorithm. Each iteration of a multiplicative weights algorithm uses $O(1/\epsilon^2)$ interventions. The two calls to multiplicative weights do a total of $O(\log n)$ iterations, and hence $O((\log n)/\epsilon^2)$ interventions.

The clean-up steps (steps 5-11) take $O(\log(1/\delta_1)/\epsilon^2)$ interventions for each node in $M_2$, and therefore $O(|M_2| \log(|M_2|)/\epsilon^2) = O((\log \log n)(\log \log \log n)/\epsilon^2) = O((\log n)/\epsilon^2)$ interventions in total.

$\square$

Finally, in Algorithm 8 we show how to boost the error from Algorithm 7 by repetition.

---

**Algorithm 8** Noisy Central Node Algorithm. Finds root with probability $\geq 1 - \delta$.

---

**input** Observational tree $G_0$. Confidence parameter $\delta$.

1:  $\delta_0 \leftarrow \delta/(4n)$.
2: **while** true **do**
3:     $v \leftarrow$ Algorithm 7.
4:     Let $u_1, \ldots, u_d$ be the neighbors of $v$.
5:     Do $\lceil 2 \log(1/\delta_0)/\epsilon^2 \rceil$ interventions $\mathrm{do}(v = 1)$ to get estimators $\hat{a}^{v:u_i}$ of $1(R \in B_G^{v:u_i})$ with soundness parameter $\delta_0$.
6:     **if** $\hat{a}^{v:u_i} = 0$ for all $i \in [d]$ **then**
7:         Output $v$.
8:     **end if**
9: **end while**

---

**Theorem 5** (Restatement of Theorem 4). *Algorithm 8 uses $O(\log(n/\delta)/\epsilon^2)$ queries in expectation, and returns $R$ with probability $\geq 1 - \delta$.*

*Proof.* **Correctness** The maximum degree of a node is at most $n$. Thus, by a union bound, on any given iteration the estimators $\hat{a}^{v:u_i}$ are all correct with probability $\geq 1 - \delta/4$. So if on a given iteration $v = R$, then the algorithm returns $v$ on that iteration with probability $\geq 1 - \delta/4$. Otherwise, if $v \neq R$, then the algorithm returns $v$ with probability $\leq \delta/4$. Therefore, by the error-bound of Algorithm 7 (Proposition 7) the probability of outputting a true positive on any given iteration is $\geq (2/3) \cdot (1 - \delta/4) \geq 1/2$, and the probability of outputting a false positive is $\leq (1/3) \cdot (\delta/4) \leq \delta/12$. Overall, the probability of encountering a true positive before a false positive is therefore $\geq (1/2)/(1/2 + \delta/12) \geq 1 - \delta$. So the algorithm outputs the correct root with probability $\geq 1 - \delta$.

**Query complexity** Since the probability of finding a true positive on any given iteration is $\geq 1/2$, the expected number of iterations $N$ is at most 2. On each iteration, the call to Algorithm 7 incurs a cost of $\leq Q_1 = O((\log n)/\epsilon^2)$ queries, and the interventions from step 5 incur a cost of $\leq Q_2 = O(\log(n/\delta)/\epsilon^2)$ queries. So the expected number of interventions is $\leq \mathbb{E}[N \cdot (Q_1 + Q_2)] \leq 2(Q_1 + Q_2) = O(\log(n/\delta)/\epsilon^2)$.

$\square$

Figure 6: **Different skeleton selection strategies, similar qualitative behaviour:** Average number of interventions for finite sample algorithms for small graphs, $n = 10$ (left column) and $n = 20$ (right column), where each row is a skeleton generation strategy: a) random trees, b) random power trees, c) random graphs + BFS tree, for varying ranges of $\epsilon$.

**Remark 2.** *The final post-processing step in Algorithm 8 and can also be applied to the result of Emamjomeh-Zadeh et al. (2016) to fix their algorithm's suboptimal dependence with respect to $\delta$.*

## I Complete empirical results

We consider several possible experimental settings and for each setting we simulate 200 random trees of $n$ nodes with randomly generated parameters. We generate the undirected tree with three different strategies: a) sampling uniformly from the space of undirected trees with a randomly generated Prüfer sequence, b) generating power-law trees, and c) generating high degree $d = n/2$ random graphs, sampled asymptotically uniformly in the space of graphs, and then creating an undirected version of the BFS. For each of these strategies, we use the NetworkX library functions: a) `random_tree`, b) `random_powerlaw_tree`, c) `random_regular_graph`. Once we have an undirected tree, we pick the root node uniformly at random among the nodes, and orient the edges accordingly.

Figure 7: **The gain of using central node increases with the size of the graph:** Average number of interventions for finite sample algorithms for $n = 10$ strategy a (top left), $n = 30$ strategy a (top right), $n = 50$ strategy b (bottom left), $n = 1000$ strategy a (bottom right). In this last experiment information greedy is too computationally expensive, so we omit it.

For most of the experiments we focus on binary random variables, so when not explicitly mentioned the results are for binary variables. In the end of the Section in Figure 10, we show the same conclusions generalize also to the discrete case.

In the binary random variable case, each random variable is a function of its parent: if $X_{Pa_i} = 0$, then $X_i \sim \text{Bern}(\epsilon)$, else $X_i \sim \text{Bern}(1 - \epsilon)$, where for each random variable we sample $\epsilon$ uniformly from a range $[\delta, 0.5 - \delta]$. The root node is distributed as $X_r \sim \text{Bern}(0.5)$.

For discrete variables, we focus on the three valued case. Each random variable takes value $x \in \{0, 1, 2\}$ and agrees in value with its parent with probability $\epsilon$, while it takes the other two values each with probability $(1 - \epsilon)/2$, where for each random variable we sample $\epsilon$ uniformly from a range $[\delta, \frac{1}{3} - \delta]$. The root node $X_r$ has a uniform probability of taking each value.

We compare three finite sample algorithms, all using the posterior update described in Appendix B: a baseline algorithm that intervenes on a node randomly selected using the probability of being root in the current prior, the information greedy algorithm, implemented following the sampling strategy presented in Appendix C.2 with $N = 50$, and our central node algorithm presented in Algorithm 1.

Figures 6 and 7 show the average number of interventions required to find the root node, for different values of $\delta$ (or, in other words different ranges in which we sample for each random variable $\epsilon$). In all cases, as the $\epsilon$ get closer, it becomes easier to identify the root node. Figure 6 shows that for different types of trees, generated with the three strategies described before, the baseline and information greedy algorithm are consistently outperformed by our central node algorithm. Figure 7 shows that when the size of the graph increases, so does the gain of central node w.r.t. the baseline algorithm. Moreover, the information greedy algorithm is much more computationally expensive than the central node, because of all the simulated interventions required in the sampling strategy explained in Appendix C.2, making it unfeasible to run at $n = 100$.

Figure 8: **Average number of interventions decreases with the number of available samples per intervention:** Average number of interventions for the finite sample extension of the noiseless central node algorithm for $n = 10$ strategy a (top left), $n = 10$ strategy b (top right), $n = 50$ strategy a (bottom left), $n = 50$ strategy c (bottom right), plotted in the number of samples per intervention for various ranges of $\epsilon$. At 50 samples per intervention in most settings (especially the easier setting where the different values of $\epsilon$ of each random variable are all very similar $\sim [0.24, 0.26]$) the simple finite sample extension of the noiseless central node algorithm reaches similar performance to the noiseless case, thus justifying this simple extension.

Figure 8 shows the behavior of the finite sample extension of the noiseless central node algorithm presented Algorithm 2, in which the algorithm continues getting more interventional sample for a given intervention target until it reaches a certain confidence in which branch the root node lies, or until its sample budget for that intervention is reached. In Figure 8 we show the average number of interventions required to find the root node for different settings of $n$ and different tree generating strategies, when varying the number of interventional samples collected for each intervention from 1 (Algorithm 1) to 50. Each curve represented different values of $\delta$. As expected, the behavior of the central node algorithm improves smoothly with the number of interventional samples, quickly converging to the performance of the noiseless Algorithm 2. Figure 9 shows the performance of the $K$-central node algorithm for varying $K$, $n$ and tree generation strategies.

Figure 9: **Different settings, same behaviour for K central node algorithm for various K:** Average number of interventions for finite sample algorithms for $n = 10$ strategy a (top left), $n = 10$ strategy b (top right), $n = 20$ strategy c (bottom left), $n = 50$ strategy a (bottom right).

Figure 10: **Results for discrete random variables:** Average number of interventions for finite sample algorithms for small graphs, $n = 10$ (left column) and $n = 20$ (right column), where each row is a skeleton generation strategy: a) random trees, b) random power trees, c) random graphs + BFS tree, for varying ranges of $\epsilon$.