[Reviews · NeurIPS 2019]

Reviewer 1



The specific topic of this paper is somewhat outside my area. Originality: the paper seems original/novel, as far as I can tell. Quality: the algorithms and theoretical results seem interesting, at least from a theoretical point of view. The details of the proofs are a bit outside my area of competence so I won't comment on those. The experiments seem sound. Clarity: the paper is quite clearly written. Significance: I am less confident about the broad significance of the work because I am not so familiar with work which focuses on tree structures (see below). However, the introduction of the general adaptive strategy seems like it would be of broad interest. One element that is lacking in my opinion is the motivation for this restricted setting. I understand the general motivation for optimal experiment planning for causal discovery, but I'm not sure what justifies the restriction to tree structures. Is there some (scientific) setting where this is substantively reasonable? If so, could there be a real data application from such a scientific setting, or at least a simulation that closely emulates that setting? - (typo) pg 3 ln 120: {X1,X4,X5}

Reviewer 2



The authors proposed a suite of algorithms for learning the structure of the causal graph under different assumptions (infinite and finite interventional sample, single vs. K intervention, non-manipulable variables). The assumption about the type of underlying causal graphs is quite stringent: a tree with no v-structure. Authors do not provide a compelling real-world example where this assumption makes sense. Nevertheless, this work seems to provide a theoretical insight to the very specific class of problems. Overall the paper is written clearly for readers to follow without any interruptions in general (there are some issues with how the paper is organized and I will talk about this below.) Theorems and their proofs seem clear to me. The impact of theoretical results seems minimal due to its lack of applicability. Lemma 1 suggests that the ratio between P(Y=y)/P(Y=y|X_i=1) plays an important role to update one’s belief on where the root of the given tree is. Regardless, the central node algorithm (a finite sample case, Algorithm 1) does not make use of such information. Then, the algorithm might not be optimal with respect to the number of total interventional samples (not the number of interventions under noiseless setting). Consider a case where a node X_i is selected by the algorithm based on a centrality measure where the following holds: P(Y=y) is very similar to P(Y=y|X_i=1), in other words, the strength of edges between X_i and all of its neighbors are too weak. In such case, interventional samples will not be able to change the posterior distribution sufficiently compared to samples obtained by intervening on other variables (close to the central node but associated with larger ratios) will. This case suggests that the ratio should be incorporated into a centrality measure in a finite sample case. Although you have shown the sub-optimality of an information greedy algorithm with respect to the number of interventions under noiseless setting, you didn’t show how such an information greedy algorithm works w.r.t. finite-sample case. Section 3.4 partially addresses the problem by defining epsilon, and the number of samples we need to pull under the intervention on a selected node X_i to confidently determine where the root is. (what are estimators “a” in Proposition 2? You didn’t define it.) Q: Is there a way to incorporate a prior distribution, e.g., Dirichlet distribution, over variables? Is it unnecessary? ========== - after response: In the finite case, not only weak edges can be left out but also non-edges can be included (false positives). I am keeping my initial score of the paper, which is a weak acceptance.

Reviewer 3



This work proposes adaptive algorithms for selecting nodes to intervene in order to discover the causal directions. Their approach is based on an assumption that the underlying structure is a tree. They proved that their algorithms could identify the causal structure with number of interventions within a constant factor of the optimal algorithm. Their analysis are divided into noiseless and noisy observation settings. They also provide algorithms for settings in which there exists intervention restriction (some nodes cannot be intervened) or settings in which it is ideal for the experimenter to perform k interventions in sequence. The paper is sound and nicely written. Although, I have not gone through the proofs in the appendix. A precise definition of tree would make the claim of the paper more rigorous. Based on the presented algorithms in the paper, it seems that the underlying assumption is that the true causal structure should be a directed tree (it only has one root). If this is the case, it should be mentioned in Section 2. As it is also mention in the Introduction section, the assumption that the underlying structure is a tree is quite restrictive. Could the authors elaborate on different scenarios that such tree-structures may emerge, i.e., for what type of applications such tree-structures are relevant? The definition and existence of the central node highly depend on the tree-structure of the graph. This may limit the generalizability of the proposed algorithms.

Reviewer 4



The authors study the problem of intervention design for learning tree causal structures. The proposed approach is adaptive, in which at each step a Bayesian prior over the underlying structures is updated and the next intervention target is chosen based on the updated prior. The authors first assume that infinite observational and interventional data are available, and then provide the following extensions: 1. Specific set of nodes cannot be intervened on 2. K interventions are designed simultaneously 3. finite interventional data The assumptions of the work are the following: - Infinitely many observational samples are available. - There are no latent confounders and no selection bias. - Underlying causal structure is an undirected tree. (Therefore, the causal model can be specified completely by identifying of the root vertex). - Causal Markov and faithfulness assumptions. The authors assume that the underlying structure is a tree with no v-structures (otherwise the graph can be decomposed into smaller subgraphs with this property). Although reasonable in theory, there is a major issue with this assumption. In the case that the essential graph graph is tree, it is highly unlikely to have large chain components in the essential graph. In fact, checking a random directed acyclic tree, we see that majority of the edges will be learned in the essential graph and the chain components are usually of very small sizes compared to the whole essential graph. Therefore, chain components of size, say n=100, usually never occur in the essential graph. Therefore, it seems necessary that the authors provide a strategy to deal with the case that we have several separate tree chain components (not just one component). It seems that the main contribution of this work is Lemma 1 and connection of this lemma to existing results, which is explained in lines 141-143. But the intuition behind lemma 1 and the explanation in lines 141-143 is not quite clear. I would appreciate it if the authors elaborate these parts. The interventions are assumed to be single target. Is there any straight forward way to extend the updates in Lemma 1 to simulations interventions? Following Lemma 1, the authors say that "This result implies that the only relevant interventional values are those of the neighbors of the intervened node. This is a critical observation that informs the development of our approach." This statement seems obvious, since having observational essential graph, an interventional essential graph is only dependent on the neighbors of the intervened variable. See [Hauser and Buhlmann, "Characterization and greedy learning of interventional Markov equivalence classes of directed acyclic graphs," 2012]. The type of interventions used in this work is Pearl's atomic intervention, in which the value of the intervened variable is set to a certain value. The issue with this type of intervention is that it may be possible that the cause affects the effect only under certain outcomes. Therefore, the specific outcome tested under the atomic intervention may not influence the effect. (for instance, consider a switch that only for certain values activates another variable). Due to this fact, is there another assumption regarding the influence of the atomic interventions needed in this work? Should condition 1 also contain the case for do(X_i=0)? In line 256, the authors state that "For simplicity, we restrict our discussion to the case in which the X_i are binary variables, although our techniques may be applied to more general settings as well." Unfortunately, the extension is not clear. I would appreciate it if the authors clarify how the extension should be done. There are similarities between the main idea in this work and [Ghassami et al. "Optimal Experiment Design for Causal Discovery from Fixed Number of Experiments" 2017], which is also on tree structures. For instance, here, the main idea is finding a central node (Definition 1) and there, the main idea is finding a separator node (Definition 8), which have the same definition. A comparison seems necessary, of course the setting in that paper is non-adaptive. Experiments: In the experiments, the authors have restricted themselves to binary variables. It seems necessary to evaluate the results for other type of variables as well. There are no real comparison of the results with other approaches. Perhaps the most relevant work is [Hauser and Buhlmann, "Two optimal strategies for active learning of causal models from interventional data," 2014] which also considers an adaptive setup. the authors can comparing required number of interventions. Also, the reason for high interest in the literature towards non-adaptive setup is that an adaptive setup can be viewed as a non-adaptive, in which we only design one intervention. Therefore, one can compare one step of an adaptive setup with a non-adaptive with intervention size 1.

[Author Response · NeurIPS 2019]

We thank the reviewers for their insightful comments and interest in the work, find below line by line responses. For
space reasons we do not reply to comments on organization or typos but have incorporated them into the paper.

**R1, R2, R3, R5: Limitation to trees.** Several reviewers raised questions about our limitation to tree graphs. We
apologize for not motivating our assumption more clearly, and will modify the paper to clarify that the algorithm works
whenever the undirected components of the graph form a forest. As briefly mentioned in L51-52, after the observational
stage, the undirected components of the essential graph are chordal. Orienting each of them provides no information on
the others, thus they have to be handled separately by any algorithm. Our algorithm only requires these components
be trees, but they do not need to be connected. Note we can identify all v-structures from observational data. For
example this assumption is satisfied when the original graph is bi-partite, since chordal components of bi-partite graphs
are forests. Examples of bi-partite causal graphs occur in system biology networks, e.g. gene-disease networks or
gene-protein networks [https://www.ncbi.nlm.nih.gov/pubmed/27265032]. We will add this discussion to the paper.

**R2: ...information greedy algorithm is suboptimal. (low significance)** While we don't dispute this, we point out
that the dramatic failure of information greedy puts our problem in a different class from many active learning scenarios.

**[the case in which]** $P(Y = y)$ **is very similar to** $P(Y = y|X_i = 1)$**.** As R2 correctly states, our algorithm may not
be optimal if some edges are much weaker than others (i.e. when Condition 1 in Sec. 3.4 does not hold). Note that
under Condition 1, we provide formal guarantees for a version of our algorithm. We also believe that Condition 1 is
justified in real-world examples: this is because with finite observational data, we expect weak edges to be left out by
methods that learn the graph skeleton.

**...how such an information greedy algorithm works w.r.t. finite-sample case.** In the finite sample case, the
information greedy algorithm must take more interventions than in the infinite sample case. Hence the number of
required interventions is still linear in the size of the graph $n$ for our counter example structure. Meanwhile under
Condition 1, the infinite-sample central node algorithm can be applied to the finite sample case by repeating each
intervention until a branch has at least $1 - \delta/(\log n)$ probability, which requires order $\log \log n$ repetitions in expectation
by Prop. 2. By the union bound then, this algorithm finds the solution with $1 - \delta$ confidence in number of interventions
nearly logarithmic in $n$, a major speedup over information greedy.

**...incorporate a prior distribution...** We only consider priors over the graph structure. Our approximation guarantees
with respect to the optimal algorithm hold for any prior assumed over the source node locations (see Sec. 3.1).

**R5: interventions are assumed to be single target. Is there any straightforward way to extend the updates...to**
**simulations interventions?** Lemma 1 can indeed be extended, we did not originally include it because it would
complicate the paper and raise several corner cases which complicate the algorithms. We will add it to the supplement.

**The type of interventions used in this work is Pearl's atomic intervention...it may be possible that the cause**
**affects the effect only under certain outcomes.** We wrote the paper with $do(X_i = 1)$ for notational simplicity, as
long as there is any value $a_i$ for which $do(X_i = a_i)$ affects the effect variables, we can find it from the observational
data and the theory will still hold. We will expand and rigorize this discussion in the camera-ready.

**the extension is not clear...In the experiments, the authors have restricted themselves to binary variables.** We
will add a discussion and new experiments for the non-binary variables, the results are qualitatively the same. Essentially,
the extension hinges on the fact that we simply need the result of the do-intervention to provide information on the
direction of the root via the posterior update (Lemma 1), all our results follow from this.

**Connection with [Ghassami et al. 2017]:** Thank you for pointing us to this paper, we will be sure to cite it and make
the connection clear. Indeed, their Definition 8 and our central node definition are the same, and our algorithm can be
seen as repeating the first step of ProBAL with evolving posteriors. In fact, this is true for any non-adaptive algorithm
whose first step is intervening on the central node of the prior. However, ProBAL is a non-adaptive algorithm designed
for a fixed budget setting (number of interventions). Interestingly, our results show that it is a good algorithm with
a constant factor approximation guarantee even in the adaptive, fixed confidence noiseless setting. We also show an
adaptation of this works well in case of restricted nodes and in the noisy case with an assumed bound on the noise level.

**Comparison to [Hauser and Buhlmann, 2014]:** While OptSingle is non-adaptive, it can be turned into an adaptive
algorithm if one applies it repeatedly on the remaining essential graph after the intervention and application of Meek
rules. The objective is to minimize the number of unlearned edges in the worst case graph. In the case of a tree with
a uniform prior, this algorithm will also intervene on the central node, since only that will minimize the worst case
number of unoriented edges. However, when the prior is non-uniform the two algorithms are not equivalent: OptSingle
will only use the structural properties and not take into account the prior. In the noisy case, a single intervention will not
change the support of the posterior distribution, hence OptSingle will continue intervening on the same node without
making any progress. Hence we did not see a straightforward way to compare to it in our experiments, all of which
consider the noisy case. We will add appropriate citations and discussion of these papers in the camera-ready.

[Meta-Review · NeurIPS 2019]

As pointed out by the reviewers, these are the strengths and weaknesses of the paper: STRENGTHS The paper proposes algorithms for learning causal trees with intervention data under various assumptions, including infinite observational and interventional data, finite interventional data, allowing K interventions, and limiting the tree nodes that can be intervened on. There is a theoretical analysis on the bounds for the number of required interventions. The paper is overall clearly written. FOR IMPROVEMENT The main concern about this paper is the applicability of the proposed algorithms since they focus only on very specific type of causal graphs (causal trees with no v-structure). The authors should discuss the significance of being able to learn such graphs. Other points that should be discussed are the possibility of having several tree components, possible extensions to multinomials, and comparison with competing methods.